# Externalized Reusable Permanent Pacemaker for Prolonged Temporary Cardiac Pacing in Critical Cardiac Care Units: An Observational Monocentric Retrospective Study

**DOI:** 10.3390/jcm11237206

**Published:** 2022-12-04

**Authors:** Maxime Beneyto, Matthieu Seguret, Marine Taranzano, Pierre Mondoly, Caroline Biendel, Anne Rollin, Fanny Bounes, Meyer Elbaz, Philippe Maury, Clément Delmas

**Affiliations:** 1Electrophysiology and Pacing Department, Rangueil University Hospital, 31059 Toulouse, France; 2Intensive Cardiac Care Unit, Rangueil University Hospital, 31059 Toulouse, France; 3INSERM U1297, Paul Sabatier University, 31059 Toulouse, France; 4Anaesthesiology and Critical Care Unit, Toulouse University Hospital, 31400 Toulouse, France; 5REICATRA, Institut Saint Jacques, 31059 Toulouse, France

**Keywords:** temporary pacing, pacemaker, reversible conduction disorder, acute cardiac care

## Abstract

Introduction: The use of temporary cardiac pacing is frequent in critical care units for severe bradycardia or electrical storm, but may be associated with frequent and potentially severe complications, especially when indwelling for several days. In some cases, transient indication or ongoing contraindication for a permanent pacemaker justifies prolonged temporary pacing. In that case, the implantation of an active-fixation lead connected to an externalized pacemaker represents a valuable option to increase safety and patient comfort. Yet, evidence remains scarce. We aimed to describe the population receiving prolonged temporary cardiac pacing (PTCP) and their outcomes. Methods: We retrospectively included all consecutive patients, admitted to our hospital from 2016 to 2021, who underwent PTCP. We collected in-hospital and six-month outcomes. Results: Forty-six patients (median age of 73, 63% male) were included, and twenty-nine (63%) had prior heart disease. Indications for PTCP were found: seventeen (37%) potentially reversible high-grade conduction disorders, fourteen (30%) indications for permanent pacemaker but ongoing infection, seven (15%) cardiac implantable electronic device infections requiring extraction in pacing-dependent patients, seven (15%) severe vagal hyperreactivity in prolonged critical care hospitalizations, and one (2%) recurrent sustained ventricular tachycardia requiring overdrive pacing. The median PTCP duration was nine (5–13) days. Ten (22%) patients exhibited at least one complication during hospitalization. Twenty-six (56.5%) patients required definite device implantation (twenty-five pacemakers and one cardioverter-defibrillator) and twenty (43.5%) did not (fifteen PTCP device removal for recovery and five deaths under PTCP). At six months, two (5%) deaths and two (5%) new infections of a definite implanted device occurred, all in patients with initial active infection. Conclusion: The use of prolonged temporary cardiac pacing, with an active -fixation lead connected to an externalized pacemaker, is possible and reasonable; this would allow for the possible recovery or resolution of contraindication for definite device implantation.

## 1. Introduction

The use of temporary pacing wires is the traditional way of providing temporary internal cardiac pacing for several hours, up to several days. Its main function is bridging the transfer to permanent pacemaker implantation in a patient with new high-grade AV block, but it may be used in any situation involving bradycardia or the need for overdrive pacing; no contraindication exists [1]. The role of temporary pacing wires has been shaped by clinical practice rather than clinical trials [2], and all recommendations on their use have a level of evidence C in the European guidelines [3]. Yet, the shortcomings of temporary pacing wires are well known: lead dislodgement, associated with undersensing, loss of capture and arrhythmogenicity, infection (especially when indwelling over 48 h), venous thrombosis, cardiac perforation, patient mobility restriction, and patient rehabilitation limitation [4,5]. These complications may especially arise in situations requiring longer temporary pacing: (1) expected transient conduction disorder or sinus bradycardia, (2) electrical storm requiring overdrive pacing, (3) cardiac implantable electronic device (CIED) infection requiring extraction in a pacing-dependent patient, and (4) ongoing infection contraindicating direct CIED implantation in a pacing-dependent patient.

In these cases and to circumvent the limitations of temporary pacing wires, the use of an externalized permanent pacemaker generator, connected to a transcutaneous active-fixation lead, has been proposed [6,7,8]. This prolonged temporary cardiac pacing (PTCP) method appears safer and more comfortable for the patients. Therefore, it allows patient mobilization and transfer to a ward equipped with telemetry. Additionally, PTCP has been shown to save costs compared to temporary pacing wires when used for at least 18 h [9]. Finally, PTCP has entered the latest European guidelines, but only with a low level of evidence (class IIa, level of evidence C) [3].

Available knowledge concerning PTCP use in critical care units remains limited. We aimed to describe the population receiving PTCP, its indications, and its outcomes in a large tertiary university center.

## 2. Methods

### 2.1. Study Design

We conducted a retrospective monocentric cohort study in our tertiary care regional center in France.

### 2.2. Study Population

We included all consecutive patients admitted to the intensive care or intensive cardiac care units of our hospital from June 2016 to December 2021, who underwent the implantation of an active-fixation lead connected to an externalized permanent pacemaker for PTCP. We collected medical history, cardiovascular risk factors, clinical and electrocardiographic data, ongoing treatments at PTCP implantation, PTCP indication, and PTCP implantation method. The comprehensive list of collected variables is available in Appendix A.

### 2.3. Complications and Outcomes

All complications and outcomes were reviewed from patients’ medical records.

We collected common complications associated with the pacemaker lead implantation: pneumothorax, tamponade, bleeding, lead dislodgement, and bacteremia. We also collected ventricular arrhythmias, as they are the adverse events described with temporary pacing wires, ref. [4] and supraventricular arrhythmias.

We considered three possible in-hospital outcomes: definitive device implantation, mere PTCP removal, or death before definitive implantation or PTCP removal.

At 6 months, we collected the occurrence of definitive device implantation in patients who had PTCP removal, CIED infection in patients implanted with a definitive device, and death.

### 2.4. Prolonged Temporary Cardiac Pacing

The implantation of the PTCP devices was performed by an electrophysiologist in an operating room dedicated to CIED implantation. The procedures were performed under local anesthesia unless the patient was already under deep sedation.

After adequate skin preparation, a percutaneous venous puncture (usually of jugular or subclavian vein) was performed and an introducer sheath was inserted using Seldinger’s technique. Under fluoroscopy guidance, an active-fixation lead was implanted in the right ventricle with the help of a curved stylet. Once the desired lead positioning was obtained and stable, the lead was screwed into the myocardium and the reliability of its fixation was assessed by gentle pushing. If sensitivity and threshold testing were satisfactory, the insertion sheath was removed and the suture sleeve pushed along the lead, up to the puncture site. The lead was finally secured by fixing the sleeve to the skin using a non-absorbable braided suture. Then, a sterilized reusable permanent single-chamber pacemaker generator was connected to the lead and taped to the patient’s skin (Figure 1). A light compression dressing was applied to the venous puncture site for 24 h. Sterile dressing care was performed daily thereafter.

The procedures did not require antibioprophylaxis. Antiplatelet therapy was maintained. If possible, anticoagulation was interrupted 4 h before the procedure and resumed the following day.

### 2.5. Ethics

According to French regulations, retrospective studies using usual care data do not need approval from an ethics committee, but have to comply with the methodology of the Comission Nationale de l’Informatique et des Libertés (CNIL). Per the General Data Protection Regulation, this study was registered at the Toulouse University Hospital (RnIPH 2022-65) and covered by the MR-004 (CNIL number: 2206723 v 0). Patients were informed that their data would be anonymously used for this study.

### 2.6. Statistical Analysis

Categorical variables were presented as counts (proportion) and continuous variables as median (interquartile range). When data were missing, their count and proportion were displayed. Fisher’s exact tests were used to compare proportions across groups and Wilcoxon’s rank sum tests for continuous variables. After ensuring their conditions of application were respected, the association of two continuous variables was assessed with linear regression and the association of a dichotomic and a continuous variable with logistic regression. A *p*-value < 0.05 was considered statistically significant. All tests were performed using R version 4.2.1 (R Foundation for Statistical Computing, Vienna, Austria).

## 3. Results

### 3.1. Population

We included 46 patients in this study. The “median patient” was a 73-year-old male admitted to a critical care unit, who had prior heart disease and a Simplified Acute Physiologic Score II of 37. Comorbidities and cardiovascular risk factors were common. Detailed characteristics are available in Table 1.

### 3.2. Indications and Devices

PTCP indications were led by potentially transient high-grade conduction disorder (17 patients, 37%) and permanent pacemaker indications in patients with a temporary contraindication to implantation due to an ongoing infection (14 patients, 30%: mainly pneumonia, urinary tract infections, and endocarditis). In total, seven (15%) patients were pacing-dependent and presented with CIED infection requiring its extraction; seven (15%) patients had severe nursing-associated vagal bradycardia inducing hemodynamic instability; and one patient (2%) had recurrent sustained ventricular tachycardia requiring overdrive pacing (Figure 2).

Population characteristics according to PTCP indications are presented in Appendix A. Patients with pacemaker indications but ongoing infection, presented with prior heart disease more often than patients with other indications for PTCP (*p* < 0.05).

PTCP was initiated 4 (2–10) days after hospital admission and consisted of a single-lead VVI pacemaker in all patients. The preferred access site was the jugular vein (Table 2).

### 3.3. In-Hospital Outcomes

PTCP lasted for a median duration of 9 (5–15) days and ranged from 1 to 63 days. The Simplified Acute Physiologic Score II at admission was not associated with PTCP duration (*p* = 0.60).

Ten (22%) patients exhibited at least one complication during their hospital stay. Two (4%) patients had a new onset of supraventricular arrhythmia, four (9%) displayed sustained ventricular arrhythmias, two (4%) experienced a lead displacement, two (4%) a bleeding (none requiring hemostatic surgery), one (2%) a bacteremia, and one (2%) a pneumothorax. No tamponade was observed. Additional information about patients developing arrhythmias during PTCP is given in Appendix A.

Twenty-six patients (56%) were eventually implanted with a definite device (nine leadless pacemakers, nine epicardial pacemakers, seven transvenous pacemakers, and one intravenous implantable cardioverter-defibrillator), fifteen (33%) had their PTCP device removed, and five (11%) died before removal or the implantation of a definitive device (Figure 3). In the subset of patients in whom PTCP was removed without the implantation of a definite device, there were fourteen males (70%), nine (45%) who had prior heart disease, with a median age of 69 (60–74) years and a median Simplified Acute Physiologic Score II of 40 (29–57). PTCP was used for a median of 9 (7–12) days before removal.

Patients under mechanical ventilation with catecholamine support, or in whom the delay from admission to PTCP implantation was longer, were more likely to die before PTCP removal or the implantation of a definitive device (*p* < 0.01, *p* < 0.01, and *p* < 0.05, respectively).

The distribution of the causes of death was the following: three treatment withdrawals or withholdals, one refractory acute respiratory distress syndrome due to COVID-19, and one brain death secondary to prolonged initial cardiac arrest.

### 3.4. Six-Month Outcomes

Among the twenty-six patients who received a definite device, two (8%) developed a CIED infection and two (8%) died within the next 6 months (Figure 2). The causes of death were multiorgan failure secondary to infection relapse in one patient, and treatment withdrawal or withholdal in a context of non-CIED-related severe infection in the other. Additional information about these two patients can be found in Appendix A.

None of the fifteen patients who did not receive a definite device required CIED implantation later on. None had died 6 months after discharge.

Longer PTCP was associated with a higher likelihood of adverse outcomes at 6 months (*p* < 0.05).

### 3.5. Outcomes According to Indications

In-hospital and 6-month outcomes according to PTCP indication are presented in Appendix A. Complication rates were similar among PTCP indications (*p* = 0.67, Appendix A).

In-hospital outcomes differed significantly according to PTCP indication (*p* < 0.0001). The highest proportion of death (2/7, 29%) occurred in patients with nursing-associated vagal bradycardia and did not exceed 12% in the other groups. All patients with pacemaker indication but ongoing infection, and all pacing-dependent patients with CIED infection, were implanted with a definitive device when they survived their hospital stay, whereas it reached only a maximum of 33% in the other groups.

The 6-month outcomes did not differ according to PTCP indication (*p* = 0.72); however, all four (100%) adverse outcomes (device infection or death) observed at 6 months occurred in patients in whom PTCP was used to delay the implantation of a permanent pacemaker until an ongoing infection was resolved.

### 3.6. Outcomes According to the Occurrence of Complications

In-hospital and 6-month outcomes did not differ between patients who experienced complications and those who did not (*p* = 0.17 and *p* > 0.99, respectively, Appendix A).

## 4. Discussion

We describe the use of an active-fixation lead connected to an externalized permanent pacemaker for PTCP in forty-six critical care patients. The main indication was that bradycardia with either a reversible cause or a temporary contraindication to definitive CIED implantation. The median PTCP duration was 9 days. In-hospital complications were not rare in this elderly and fragile population with the associated comorbidities, but the use of PTCP avoided the implantation of a permanent CIED in 43% of patients. None of the fifteen in-hospital surviving patients who did not receive a definitive device required CIED implantation or died at 6 months. The 6-month adverse events only occurred in patients for whom PTCP was used to delay the implantation of a permanent pacemaker in an infective context.

### 4.1. Indications

As previously reported [5], PTCP was used in our cohort for old, severe, and/or comorbid patients with multiple conditions, risk factors, and medications.

Many studies have been dedicated to the specific case of PTCP for CIED infections, [10,11,12,13] but this indication only came third in our study. This might explain why PTCP duration was much shorter in our study than that in Shah et al.’s study [14], but similar to that of Braun et al.’s study, which included all patients requiring PTCP [5].

Cases of nursing-associated vagal bradycardia were absent in the earliest reports on the technique, [6] but appeared in later published papers with a similar proportion as ours [14]. These complex and very specific cases are rather exceptional since most can be managed with parasympathetic antagonists or are very transient. However, when persisting and resistant, the assessment of the risk-benefit balance is challenging: potential risks of repeated syncope/asystole versus the risks inherent to prolonged endovascular pacing (infection, perforation, bleeding, and thrombosis). PTCP might appear as an elegant solution with a favorable risk profile when all other therapeutic options have failed, ignoring the conditions leading to bradycardia.

### 4.2. Complications and Mortality

We reported frequent in-hospital complications (22% of our patients) higher than in some previous reports [5,14], but we decided to be more exhaustive in associated adverse events collection. Age and sex ratio were in the same range as in previous studies, but published data do not allow us to compare patient severity to fully understand the complication excess we observed.

We observed a similar proportion of lead displacement as in the literature (0.3 up to 6.6%) [11,14], the same proportion of bacteremia as Kawata et al. (2%), [10] and more pneumothorax (2 vs. 0.6%) but less tamponade (0 vs. 1.3%) than Xiao et al. [11]. Yet, our population is small and may not accurately capture these rather rare complications.

Supraventricular arrhythmias had not been reported previously. They may have been overlooked as they are common in critical care settings, especially in patients with a preexisting heart condition [15]. The link between an external pacemaker with a ventricular lead and supraventricular arrhythmias is not obvious, yet we felt it was important to collect these events to investigate a potential association. Ventricular pacing can be responsible for the onset of supraventricular arrhythmias due to the retrograde P waves it may induce [16] in patients with preserved retrograde atrioventricular conduction (which may exist despite complete atrioventricular anterograde block). By colliding with the sinus P wave, the retrograde P wave can trigger atrial fibrillation or an atrial flutter. In our population, we recorded only two supraventricular arrhythmias. Both occurred in patients with severe infections, which are known to facilitate supraventricular arrhythmias [15], and it is probable that PTCP was little involved in these events.

Concerning ventricular arrhythmias, we reported more events than Shah et al. (9 vs. 1.5%) [14]. It is unlikely that they were a consequence of PTCP. Firstly, in all of our patients, the PTCP lead was positioned either at the apex or the mid-septum of the right ventricle under fluoroscopy. This is the standard practice in definite devices and is known to be safe [17]. Secondly, in case of lead dislodgement, the free lead may hit the walls of the right ventricle with the heartbeats and trigger mechanical premature ventricular complexes that may degenerate into ventricular arrhythmias in case of an R-on-T phenomenon. Although, in our population, none of the two patients with lead dislodgement experienced ventricular arrhythmias. Finally, after careful examination of the patient’s medical records, it appeared the underlying medical condition could largely explain the occurrence of the ventricular arrhythmias (Appendix A).

We report an in-hospital mortality rate of 10.8%, which is expected considering the severe comorbid and critical status of our patients, and similar to previous reports (10–15%) [5,13,14]. However, it is important to emphasize that none of these deaths were related to PTCP. We found that mechanical ventilation, catecholamine support, and a longer delay between admission and PTCP implantation were associated with a higher risk of in-hospital death. Even though the Simplified Acute Physiologic Score II score was not significantly associated with this risk, it is most likely that it simply denotes that the most critically ill patients are indeed more likely to die during their hospital stay and therefore during PTCP. The reasons why a longer delay before PTCP is associated with in-hospital death are more eluding. We could hypothesize that it may reflect the worse prognosis of a secondarily deteriorating medical condition, such as uncontrolled infective endocarditis creating a complete atrioventricular block.

Transient cardiac pacing, using pacing wires, has shown both a slightly higher complication rate (28%) [18] and in-hospital death rate (14%) [19]. The former seems to result from a higher risk of lead dislodgement (up to 14.1%) and a marginally higher risk of tamponade (up to 2.1%); [1] the latter may not only be secondary to the higher complication rate but also to more severe conditions, such as extended myocardial infarctions that have become less frequent nowadays [1].

### 4.3. Outcomes

The proportion of patients implanted with a definite device after PTCP was slightly lower than in previous studies (56 vs. 67%) [14]. Almost half of our cohort did not require definite device implantation, which justified the use of temporary pacing. This strategy allowed us to avoid the costs and potential complications that these superfluous implantations would have generated. The DDD-to-VVI pacemaker ratio was the same as in the study by Shah et al. (1.5), but the proportion of epicardial pacemakers was higher than in the study by Cipriano et al. (34 vs. 7.6%). Intriguingly, the use of leadless pacemakers has never been mentioned in previously published papers, even in the most recent studies. Maybe due to differences in the recruited patients, a lower proportion of implantable cardioverter-defibrillators (4 vs. 29%), and no resynchronization devices, were implanted in our study (vs. 6%) [12].

Only studies focusing on CIED infection have hitherto reported mid-term outcomes [10,11,12,13]. When accounting for this subset only, the 6-month outcomes observed in our study were roughly similar to 8% delayed CIED infections (5% at 3 months in Kawata et al.) and 8% deaths (5% at 1 year in Cipriano et al., with significant loss in follow-up).

In the whole population, we observed that all adverse outcomes at 6 months occurred in patients with the same indication. This could be explained by the fact that this group of patients combined a double condition: an ongoing infection and the need for permanent pacemaker implantation. Additionally, they were more likely to have prior heart disease than the rest (Appendix A).

It is unclear whether longer PTCP would ensure the control of the infection and thus limit the risk of definite CIED infection, or rather increase the occurrence of adverse outcomes.

### 4.4. Limitations

PTCP indications remain infrequent and despite including patients over a 6-year period, our population size remains small. Due to the retrospective and monocentric nature of this study, its results might not reflect the characteristics of all patients undergoing PTCP. Moreover, the absence of comparison with patients with the traditional temporary pacing wire or direct definite device implantation precludes a definitive conclusion; yet, the use of PTCP stems from experience with the complications associated with these older approaches. We were able to describe 6-month outcomes, which proved reassuring, but long-term data are still lacking on this topic.

The main limitation of PTCP in itself is that it requires an electrophysiologist, an operating room, and the use of fluoroscopy, which may not be available at all times or even at all, at a given hospital. Temporary pacing wires may be used until PTCP becomes possible. Furthermore, complications associated with lead implantation, such as cardiac perforation, tamponade, pneumothorax, etc., may still occur with PTCP.

## 5. Conclusions

In this retrospective monocentric cohort study, PTCP appeared reasonable whether considering in-hospital or 6-month outcomes; it also indicated that the implantation of permanent CIEDs and their potentially associated complications can be avoided in more than 40% of cases. However, its use in the context of ongoing infection is associated with a worse prognosis compared to other indications, justifying more specific attention.

## Figures and Tables

**Figure 1 jcm-11-07206-f001:**
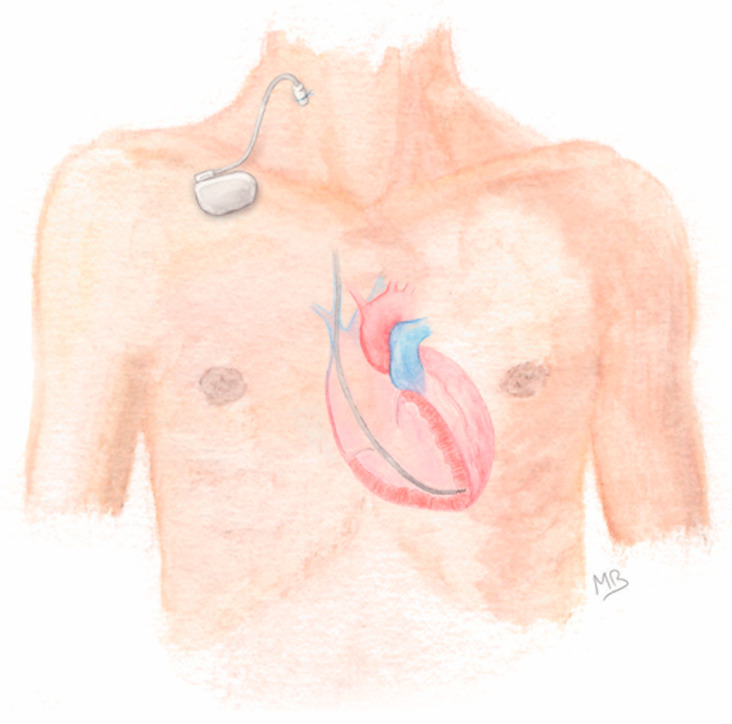
General aspect of prolonged temporary cardiac pacing (using a right jugular vein access).

**Figure 2 jcm-11-07206-f002:**
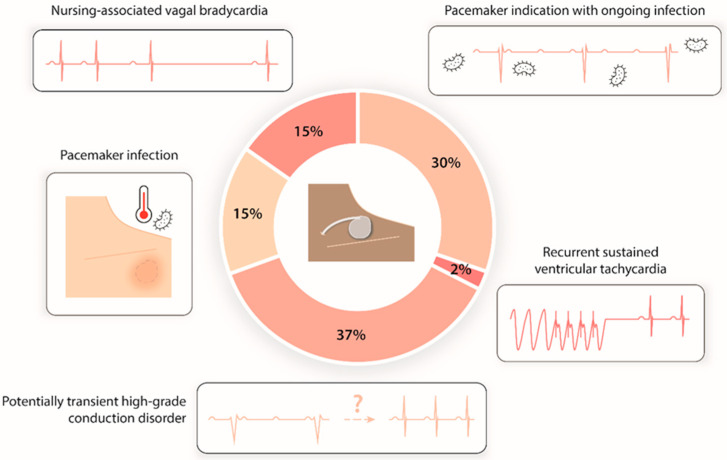
Distribution of prolonged temporary cardiac pacing indications.

**Figure 3 jcm-11-07206-f003:**
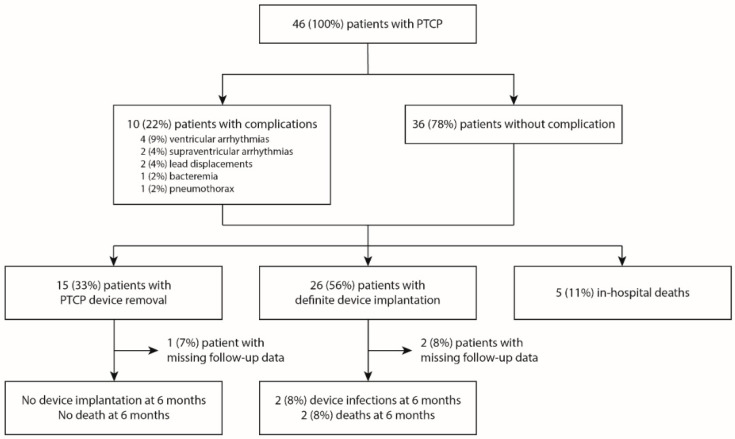
Flow chart of the population. PTCP, prolonged temporary cardiac pacing.

**Table 1 jcm-11-07206-t001:** Patient characteristics at the initiation of prolonged temporary cardiac pacing.

Characteristic	PTCP Population (n = 46)
Age (years)	73 (62–78) *
Female sex	17 (37)
Unit of admission	
Critical care unit	42 (91)
Cardiology ward	4 (9)
Cardiovascular risk factors	
Obesity	14 (30)
Diabetes mellitus	12 (26)
Arterial hypertension	21 (46)
Tobacco	18 (39)
Dyslipidemia	15 (33)
Peripheral artery disease	6 (13)
Immunosuppression	0 (0)
Respiratory insufficiency	6 (13)
Chronic kidney disease	9 (20)
Cancer	0 (0)
Prior heart disease	29 (63)
Ischemic heart disease	14 (48)
Dilated cardiomyopathy	1 (3)
Rhythm disorder	17 (59)
Valvular heart disease	13 (45)
Other	2 (7)
Medications at implantation	
Vitamin K antagonists	5 (11)
Non-vitamin K antagonist oral anticoagulation	3 (7)
Aspirin	27 (59)
P2Y_12_ receptor inhibitor	13 (28)
Unfractionated heparin	16 (35)
Low molecular weight heparin	15 (33)
Betablocker	5 (11)
Non-dihydropyridine calcium channel blocker	8 (17)
Amiodarone	9 (20)
Other anti-arrhythmic drugs	4 (9)
Catecholamines	19 (41)
Antibiotics	32 (70)
Invasive mechanical ventilation at implantation	11 (24)
Renal replacement therapy at implantation	9 (20)
SAPS II at implantation	37 (28–53) *

All characteristics are described as number (%) except for * variables described as median (interquartile range). SAPS II, Simplified Acute Physiology Score II.

**Table 2 jcm-11-07206-t002:** Device characteristics for prolonged temporary cardiac pacing.

Characteristic	
**PTCP device type**	
Single-chamber VVI pacemaker	46 (100)
**Vascular access ^a^**	
Jugular vein	23 (82)
Subclavian vein	5 (18)
**Delay from admission to PTCP implantation (days)**	4 (2–10)
**PTCP duration (days)**	9 (5–15)

^a^ Eighteen (39%) observations were missing. PTCP, prolonged temporary cardiac pacing; VVI, ventricular pacing and ventricular sensing with pacing inhibition on sensed events.

## Data Availability

Data are available upon reasonable request submitted to the corresponding author.

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
