# Peer review of "Externalized Reusable Permanent Pacemaker for Prolonged Temporary Cardiac Pacing in Critical Cardiac Care Units: An Observational Monocentric Retrospective Study"

_jcm, 2022, doi:10.3390/jcm11237206_

Round 1
Reviewer 1 Report
Beneyto and Collegues conducted a retrospective study regarding indications and complications related to prolonged temporary cardiac pacing (PTCP) in a small and fragile population admitted to intensive care or intensive cardiac care units from June 2006 to December 2021.
46 patients were included in the study, of whom 70% received antibiotics, 41% needs catecholamines, 24% inavisive mechanical ventilation and 20% renal replacement therapy at implantation.
Indications for prolonged temporary cardiac pacing were: nursing-associated vagal bradycardia 15%, pacemaker indication with ongoing infection 30%, potentially transient high-grade conduction disorder 37%, pacemaker infection 15%, recurrent sustained ventricular tachycardia.
In-hospital complications occurred in ten (22%) patients, particularly 4 patients (9%) developed sustained ventricular arrhythmias and one (2%) a bacteremia with consequent death. 26 patients were implanted with a definite device while 15 patients had their PTCP device removed. Six-month adverse events only occurred in patients for whom PTCP was used to delay the implantation of a permanent pacemaker in an infective context.
The study is interesting but some clarifications are necessary.
- Please describe in detail the population that not received a final device. Could you identify some clinical features that can predict the risk of final pacemaker implantation?
-In supplementary appendix (section Arrhythmias), authors described some cases of ventricular tachycardia. Ventricular events could be caused by placement of active fixation lead. Please clarify and comment.
-In Methods section the authors wrote that “The procedures did not require antibioprophylaxis”, but 70% of patients already received antibiotics.
How many acute infections occurred in subjects who did not receive antibiotics?
-Please discuss more extensively the sentence “In-Hospital outcomes differed significantly according to PTCP indication..” pag. 6, lines 169-170.
Could you identify some basal characteristics that may predict worse outcomes? If yes, add these findings in the results
Reviewer 2 Report
M. Beneyto and others provide an interesting case series about the prolonged temporary pacing in ICU patients, a critical topic in which randomized trials are substantially hard to realize. Even if the sample is consequently absolutely small, it appeared sufficient, and even appropriate for a full-text article.
However, I've got many issues that must be revised.
1. The overall references are scarce. In the introduction, you must provide more information about indications, contraindications, and collateral effects of usual temporary pacing, and some more information about PTCP.
2. Methods. Can you what clinical, electrocardiographic data you analyzed? It's important for reproducibility.
3. Methods. You substantially didn't expose which outcomes you'll explore, even if are reported in the results it's important to provide them in the methods (what did you expect? reproducibility?). Even many conclusions in the discussion are affected by some choices in the outcomes and in the explored variables, so more "prior" details are needed.
4. Methods. How did you correlate PTCP duration and SAPS? Did you use correlation, and association between groups? The same for PCTP duration and the likelihood of adverse events. The statistical analysis appeared insufficient, improve it. Did you use non-parametric statistics?
5. If possible, detail better complications and outcome results.
6. Discussion. Why supraventricular arrhythmias aren't reported in previous studies? Explain better why you tested them (I think it's correct and important, but can you improve this section?)
7. Discussion. You must provide data to establish a mortality of 10.8% is expected, you cannot write that without references.
8. Discussion. Even the outcomes section could be improved. More confrontation between literature and your data (expose numbers) could be useful to better understand the impact of the study.
Best regards.
Round 2
Reviewer 1 Report
The authors have improved the paper and answered all questions.
Reviewer 2 Report
I found my concerns adequally addressed and the paper improved. I recommend for publication